# Improving Parkinson's disease management through wearable technology: A cost-benefit perspective

Daniel Rodriguez-Martin[1], Andreu Català[2]*, Joan Cabestany[2]

**1** Sense4Care S.L., Cornellà de Llobregat, Spain, **2** Universitat Politècnica de Catalunya, Vilanova I la Geltrú, Spain

* andreu.catala@upc.edu

## Abstract

Parkinson's Disease (PD) is a progressive neurodegenerative disorder affecting millions of patients worldwide, with significant economic and social implications. The increasing prevalence of PD, coupled with rising healthcare costs, necessitates cost-effective solutions for patient management. Wearable medical devices, such as STAT-ON™, an inertial sensor with AI processing capability, offer an opportunity to enhance symptom monitoring, optimize therapy adjustments, and improve patient quality of life (QoL). However, their cost-effectiveness in routine clinical practice remains insufficiently explored. This study conducts a cost-benefit analysis of integrating STAT-ON™ into European healthcare systems (Spain, Sweden, Germany, Italy, and the UK). Using validated clinical data, we assess the economic impact of early symptom detection facilitated by continuous monitoring. Our findings suggest that improved symptom assessment can lead to optimized medication regimens, reducing hospitalizations and institutional care costs. While medication costs may increase, overall direct healthcare expenses decrease, leading to net savings. As an example, the study estimates potential cost reductions of up to €137.8 million in Germany and €19 million in Sweden when STAT-ON™ is used to detect advanced PD symptoms. Despite these promising results, limitations exist, including variations in healthcare costs, reimbursement policies, and real-world adoption rates across the different countries. Additionally, indirect costs such as caregiver burden were not fully analyzed. Future longitudinal studies are needed to further validate the economic and clinical benefits of wearable monitoring devices in PD management. This study highlights the potential of STAT-ON™ as a valuable tool for reducing healthcare costs and enhancing patient outcomes, supporting its adoption as a complementary solution for objective PD assessment in clinical practice.

**Data availability statement:** All relevant data are within the paper and its Supporting information files.

**Funding:** The author(s) received no specific funding for this work.

**Competing interests:** The authors have declared that no competing interests exist.

## 1. Introduction

Parkinson's Disease (PD) is a progressive neurodegenerative condition with no cure that affects approximately 7M to 10M people around the world. In Europe, the prevalence ranges from 108 to 257 per 100,000 people [1]. Age is one of the main risk factors in PD, and recent studies indicate that, due to population ageing in Western countries and increased life expectancy, the number of patients will rise exponentially, reaching 17M in 2040 [2], and 25.2M in 2050 [3]. [2] This fact, along with the high increase in the cost per PD patient, which is rated at 8.300 €/year, more than a healthy person in Europe on average [4], shows a complex economic scenario where public health systems will probably experience problems in affording care correctly. Also, according to Parkinson's Europe Association (previously EPDA), Parkinson's Disease implies a total social cost as high as €13.9 Bn per year, meaning approximately 11.600€ for patients/year as average [5]. After asking patients about their symptoms and their relation with hospitalisations, it can be found that at least 25% of PD patients have been hospitalised and that a bigger care burden is correlated to poorer quality of life (QoL) and being more susceptible to re-hospitalisation situations, increasing the cost of care of the patient [6,7].

The assessment of PD patients has always been complex, and results are not the expected by professionals and by patients. It is estimated that 40% of PD-related diagnoses are wrong in current clinical practice, and it provokes continuous frustration for the patient and the doctor [8]. Many neurologists claim that wearable sensors can be a solution for helping them to objectively monitor symptoms and provide more relevant and reliable data from the patient in home environments [9–13]. However, although the use of wearables seems promising, the cost-effectiveness of employing these medical devices in clinical routines remains poorly tested.

This article explores the use of a wearable device (the STAT-ON™ device) to illustrate how integrating such medical technology into clinical practice in PD can both reduce overall healthcare costs and improve the QoL for patients [14]. The document is structured as follows: first, it presents the characteristics and clinical validation of a commercially available and validated wearable device. It then evaluates the impact of PD on QoL, highlighting the importance of timely and accurate diagnosis of the symptoms in different stages of the disease. Then, a quick review of how therapies improve QoL in PD followed by analysis of demographic and cost data related to disease treatment across different European countries.

Finally, a model is proposed based on hypotheses that enable the estimation of treatment cost savings in five European countries—Spain, Sweden, Germany, Italy, and the UK. This model facilitates the assessment of a potential scenario in which STAT-ON™ has been successfully integrated into routine clinical practice and patient care. All the economic data are taken from already published and referenced works.

## 2. Impact of wearables on Quality of Life

This study is based on the premise that leveraging technology—specifically STAT-ON™—can enhance the QoL for individuals with Parkinson's disease without

extra expenses. STAT-ON™ provides objective data on patients' motor symptoms during daily activities, allowing neurologists to make informed treatment adjustments.

Using validated and published data, this study proposes a model for integrating STAT-ON™ into clinical practice to improve symptom assessment, optimise therapy adjustments, and ultimately enhance patient outcomes. As depicted in Fig 1, the neurologist prescribes STAT-ON™ for home use over a defined period to collect objective data on symptom patterns, durations, and fluctuations. The primary objective is that these data will enable the fine-tuning, optimisation, or modification of treatment, thereby improving patients' QoL.

The next section introduces STAT-ON™ as a validated medical device, discussing its functionalities, including the accuracy and validation in the assessment of the main motor aspects of PD, including ON/OFF state detection, dyskinesia assessment, and gait analysis. Additionally, the document explores the underestimation of motor symptoms in clinical practice, showing how delayed or inaccurate assessments contribute to suboptimal treatment and would increase healthcare burdens. Finally, a section about how therapies in Parkinson's are increasing QoL in PD patients is described, closing the rationale behind Fig 1.

## 2.1 Summary of STAT-ON™'s validation

The STAT-ON™ is a single, waist-worn Class IIa medical device that objectively monitors Par-kinson's disease (PD) motor symptoms in real-world settings. It autonomously detects ON/OFF states, dyskinesias, bradykinesia severity, freezing of gait (FoG), falls, and medication intake while providing gait and activity data [14].

Powered by machine learning algorithms, the device operates independently, with configuration and data retrieval managed via a healthcare professional's smartphone app. Its long battery life allows up to a week of continuous monitoring,

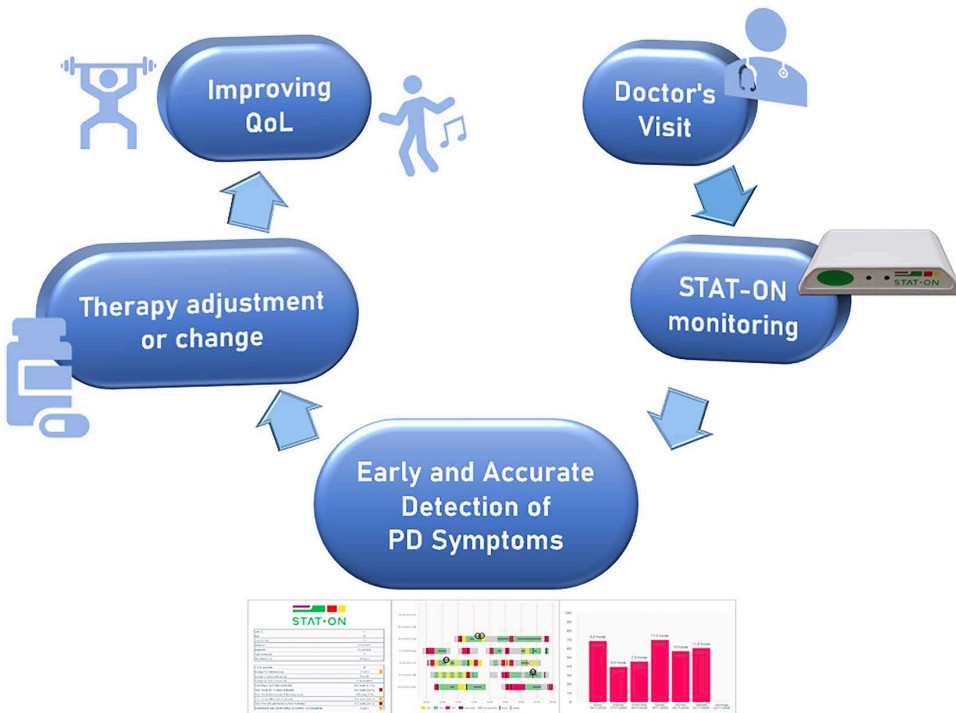

**Fig 1. Improving QoL through the use of STAT-ON™.**

with wireless charging enabling extended use. Data remains securely stored on the device, ensuring privacy, and is erased after report generation for reuse with other patients. An overview of the system is depicted in Fig 2.

The device has been validated and shown to be accurate according to published scientific evidence [14]. Some examples are the validation in the detection of bradykinesia [15], dyskinesia [16,17], FoG [17,18] or ON-OFF periods [19]. Then, the algorithms that were obtained were evaluated with new data from the clinical field. The device correlated 0.73 with the Unified Parkinson's Disease Rating Scale part III (UPDRS-III), which evaluates the motor state of the patient, in a study with 75 patients [20]. In another study, a sensitivity and specificity of 0.96 and 0.94 on the detection of motor fluctuations were obtained with 15 new patients [21]. In a clinical study with 41 patients, it was proposed to use a diary and to call the patient every hour to check if the patient was in ON or OFF state. If some outcome didn't match, then it was removed. The sensor obtained a 0.92 on accuracy with this rigorous gold-standard method [22].

The STAT-ON™ device has received external clinical validation [23–31], proving useful in several ways: it supports physicians in optimizing treatment plans, assists in identifying patients who are suitable for second-line therapies, such as deep brain stimulation (DBS), continuous dopaminergic infusions (like apomorphine or levodopa-carbidopa intestinal gel), or other advanced treatments typically considered when first-line medications no longer provide adequate symptom control [23,29,32]. The device also enhances the fine-tuning of infusion dosages [25], increases patient awareness of their symptoms [27,33], supplements or even replaces traditional patient diaries [24], and empowers patients to better recognize and understand their own Parkinson's-related symptoms [28]. The device was also tested in real conditions in 35 hospitals, showing no inferiority with other tools and proving that it could be a powerful tool to assist neurologists in hospitals [34].

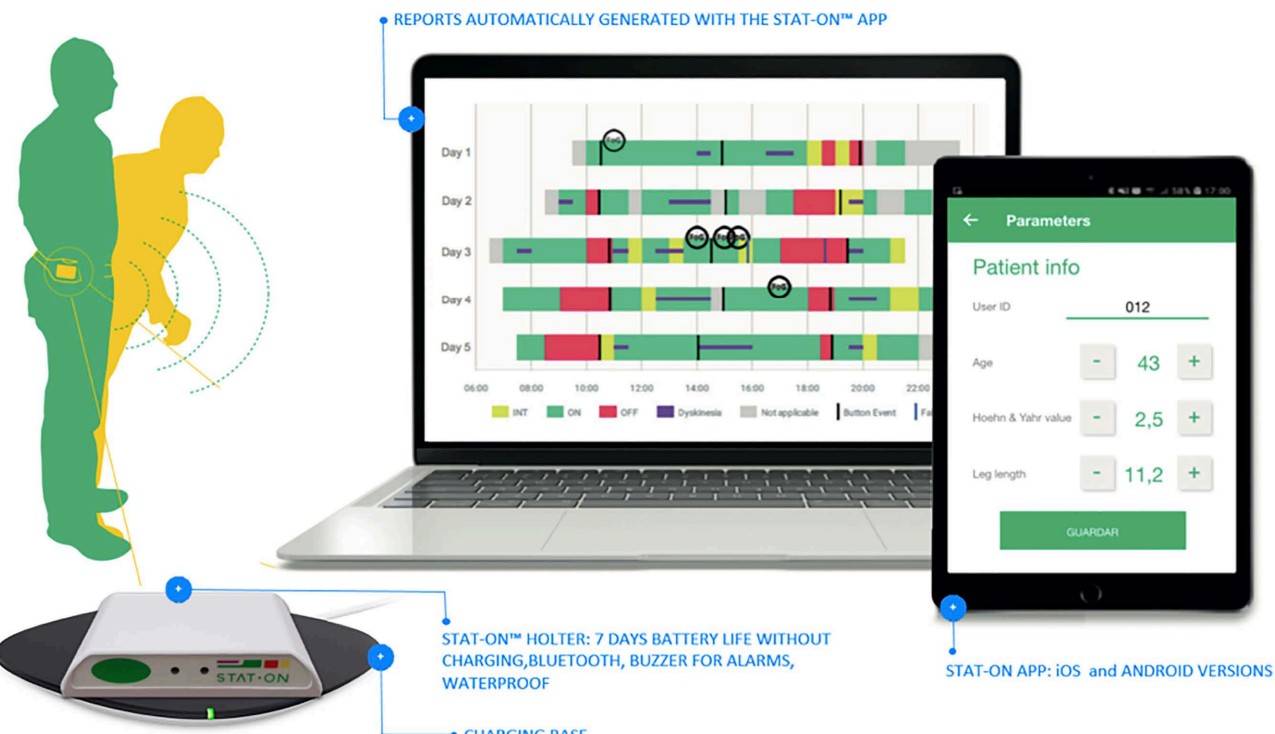

**Fig 2. STAT-ON™ general view.** The accompanying App enables configuring it at the beginning and generating the final report after the observation.

## 2.2 Underestimation of PD motor symptoms

Motor symptoms in Parkinson's disease are often underestimated in clinical practice due to subjective assessments and limited consultation time. This leads to delayed treatment adjustments and worsened patient outcomes. Traditionally, motor fluctuations are assessed using patient diaries (e.g., Hauser diaries) and validated scales such as the UPDRS [35,36]. However, these methods are subject to patient-related biases. Hauser diaries require considerable effort, leading to reduced compliance, recall bias, and patient fatigue [37]. Moreover, a recent study found a low correlation between clinician assessments and patient-reported motor states, questioning the reliability of diaries as a clinical endpoint [38].

Clinical evaluations also pose challenges, as they typically occur every 6–9 months, lasting only about 20 minutes per visit [39]. Many patients take their medication before consultations, masking symptoms and potentially biasing assessments. Additionally, the "white coat effect" and the Hawthorne effect may alter symptom presentation [40]. As a result, motor symptoms and fluctuations are often underestimated, delaying appropriate treatment adjustments.

Several studies highlight these diagnostic and therapeutic gaps. The DEEP study revealed that motor fluctuations can emerge early but are frequently overlooked in routine evaluations [41]. The DISCREPA study found that 72% of eligible patients did not receive second-line therapies, with 33.5% of those experiencing motor complications not being adequately managed [42]. The PARADISE study reported that only 15.2% of advanced PD (APD) patients received second-line therapies, often due to limited access to device-assisted therapy (DAT) or challenges in identifying suitable candidates [43]. Similarly, the global OBSERVE-PD study showed that only 43.6% of APD patients used second-line therapies, despite being treated in specialised Movement Disorders Units, which typically manage more advanced cases [44].

Delays in diagnosis and inadequate treatment significantly impact patient QoL and satisfaction. A Swedish study found that among PD patients without advanced therapies, only 9% were satisfied with their medication, dropping to 4% among APD patients [45]. Both the OBSERVE-PD and PARADISE studies reported better QoL in non-APD patients compared to those with APD, underscoring the need for earlier detection and optimized treatment strategies.

## 2.3 Improving QoL with PD therapies

The aim of this paper is to demonstrate that STAT-ON can benefit healthcare systems by reducing costs while, most importantly, enhancing patients' quality of life (QoL). Nevertheless, an objective device for monitoring motor symptoms alone cannot directly improve QoL unless its data are used to guide appropriate therapeutic interventions. To complete this rationale, it is therefore necessary to review the main therapeutic options (pharmacological or non-pharmacological) for Parkinson's disease and summarize the evidence linking them to improvements in QoL, which we present in this section [46].

Pharmacological management primarily relies on levodopa, used across most disease stages [47]. However, prolonged use can lead to motor complications such as dyskinesia, which may be mitigated by amantadine, second-line therapies [48].

Advanced symptom management follows standardized approaches [49,50]. Motor symptoms guide treatment adjustments, with COMT inhibitors (e.g., opicapone) or MAO-B inhibitors (e.g., safinamide) introduced when levodopa's effect diminishes. Dopaminergic agonists like rotigotine or apomorphine are used when symptoms become difficult to control, while duodenal levodopa infusion and Deep Brain Stimulation (DBS) are reserved for later stages [51]. Treatment selection is based on disease progression and patient profile [52,53].

Studies confirm a strong correlation between optimized PD management and improved QoL. Safinamide has shown significant improvements in UPDRS II (ADL), UPDRS III (Motor Symptoms), and UPDRS IV (Clinical Fluctuations), with a PDQ-39 reduction of −2.44 (p = 0.039) [54]. Opicapone has been shown to increase ON time and reduce OFF time (p < 0.05) in a cohort of 517 patients, with better efficacy in early-stage PD [55,56].

Rotigotine, administered via a transdermal patch, benefits patients with gastrointestinal issues and can be used as monotherapy or adjunct to levodopa. Studies confirm its efficacy in reducing UPDRS scores and improving QoL [57]. Other adjunctive therapies, such as rasagiline [58] and entacapone [59], also enhance symptom control compared to levodopa monotherapy.

For patients with difficult-to-control motor fluctuations, continuous infusion therapies provide effective alternatives. One of them is the continuous subcutaneous apomorphine infusion (CSAI), which reduces OFF time by 50–70% and improves QoL (p = 0.011) and UPDRS scores (p = 0.001) [60,61]. However, CSAI is associated with skin reactions and psychiatric side effects at high doses. Continuous duodenal levodopa infusion (CDLI) is more effective than CSAI but requires a surgical procedure [62,63]. It significantly improves wearing-off and dyskinesia, enhancing QoL [64,65]. Antonini et al. compare CSAI and CDLI, noting that while CSAI is less invasive, it is less effective for dyskinesia and often requires continued oral levodopa. CDLI, though more invasive, allows better symptom control and reduces treatment dropout through optimized dose adjustments [63].

Emerging therapies, such as inhaled levodopa [66], levodopa–entacapone–carbidopa intestinal gel infusion [67], and continuous subcutaneous foslevodopa-foscarbidopa [68], show promise but require further validation.

As a non-pharmacological approach, DBS significantly reduces OFF time [69]. However, while it improves motor fluctuations, dyskinesia, and daily activities, its long-term QoL benefits are limited due to potential speech impairments [70].

A comparative analysis by Antonini et al. [71] concluded that while DBS and CDLI provide superior symptom relief, CDLI offers greater QoL improvement. Overall, CDLI, CSAI, and DBS outperform oral treatments in managing motor symptoms.

In addition to pharmacological interventions, non-pharmacological therapies are essential components of quality-of-life-oriented care in Parkinson's disease. Physiotherapy and structured exercise programs have demonstrated benefits in mobility, balance, and fall prevention, while also contributing to improved mood and overall well-being [72–75]. Cognitive stimulation and occupational therapy can support the maintenance of functional independence [76–78], and psychological support helps patients and caregivers cope with the emotional burden of the disease [79,80]. Collectively, these approaches complement pharmacological treatment by addressing broader aspects of patient health, thereby reinforcing the importance of a comprehensive, multidisciplinary strategy for improving quality of life in Parkinson's disease.

Finally, it is also important to emphasize that therapeutic decisions should always be individualized, considering the patient's specific clinical characteristics, personal preferences, and the availability of healthcare resources [81]. This patient-centered approach ensures that both pharmacological and non-pharmacological strategies are tailored to optimize outcomes and improve quality of life. Wearable devices can benefit both pharmacological and non-pharmacological care. In addition to guiding therapy adjustments, its data on symptoms and daily activity help personalize pharmacological treatments, rehabilitation and exercise programs, reinforcing its role in comprehensive, patient-centered management.

## 3. Demographic analysis

There are several sources to quantify the PD patients in every country. Table 1 shows data of the diagnosed PD patients in several European countries, in the US and Japan, according to the five H&Y stages in 2016 [82]. It must be indicated that STAT-ON™ could be useful for all PD patients except those in H&Y 5, due to the fact that they cannot walk. This means that a total of 94.4% of patients can be well monitored with objective data using the medical device.

Patients in H&Y II and H&Y III are the most prevalent. H&Y II patients are considered to have mild symptoms with some fluctuations and are likely already taking levodopa. As their condition progresses, they will soon experience problematic fluctuations requiring additional therapies.

For the purposes of the following considerations and calculations, patients have been stratified into three groups: Mild, Moderate, and Advanced. Mild patients correspond to Hoehn and Yahr stages I–II, Moderate to stage III, and Advanced to stages IV–V. The percentage of patients in each group used for cost calculations was obtained from Table 1.

**Table 1. Number of PD patients by H&Y stage. Source: GlobalData. Parkinson's disease: global drug forecast and market analysis to 2026 [82].**

| Hoehn & Yahr stage | US | Japan | UK | Spain | France | Germany | Italy |
|---|---|---|---|---|---|---|---|
| I | 108.832 | 15.828 | 13.649 | 11.819 | 38.729 | 50.293 | 16.338 |
| II | 251.150 | 68.580 | 31.499 | 52.599 | 89.375 | 116.061 | 37.702 |
| III | 293.008 | 183.175 | 36.749 | 15.758 | 104.271 | 135.404 | 43.986 |
| IV | 139.013 | 121.674 | 17.849 | 21.082 | 50.646 | 65.768 | 21.365 |
| V | 32.709 | 53.094 | 4.200 | 5.217 | 11.917 | 15.475 | 5.027 |
| Total | 824.712 | 442.351 | 103.946 | 106.475 | 294.938 | 383.001 | 124.418 |

## 4. Cost of PD care

This section reviews the main costs of Parkinson's disease (PD), focusing on direct healthcare and medication expenses reported in different countries. Direct costs include hospitalisations, consultations, and medical equipment, while medication costs cover treatments such as levodopa and dopamine agonists. Both increase with disease progression, reflecting greater resource use in advanced stages.

Estimating the global cost of PD is complex due to differences in healthcare systems, treatment availability, and economic factors across countries [4]. Costs are generally categorised into:

• Direct medical costs (e.g., medical treatment, medication, home medical equipment)

• Direct non-medical costs (e.g., in-home assistance, alternative care, uncovered medication expenses)

• Indirect costs (e.g., productivity loss due to sick leave, early retirement, work absences for medical visits)

PD-related expenses vary significantly by country and disease stage. For instance, in Germany, patient costs range from €18,660 (H&Y I–II) to €31,660 (H&Y III–V) [83]. Similarly, in Spain, a study reporting costs on a quarterly basis showed an increase from €1,880 in the early stages to €6,266 in advanced stages over a 3-month period (Table 2). To estimate annual costs, these figures should be multiplied by four [84]. In this 4-year longitudinal study in Spain it is shown that average costs increased from €2,000 (8000€/year) to €4,000 (16,000€/ year) per patient over time. Cubo et al. report that pharmacological costs account for 34% of total direct expenses, with hospitalisations and patient care being the major contributors [85]. This last data is crucial for the calculation of costs in the following tables.

Moreover, healthcare costs fluctuate depending on national minimum wages, drug pricing, and treatment availability. Some therapies accessible in the U.S. may not be available in Europe, and vice versa.

A Swedish study in 2021 also revealed a significant cost escalation by disease stage (see Table 3). Costs rise from €5,630 in H&Y I to €95,000 in H&Y V, with a €45,000 jump from H&Y IV to V and nearly €30,000 from H&Y III to IV [86].

In addition to the European countries, the United States presents a unique context due to its large population of PD patients. Numerous economic studies have been conducted in this setting, offering valuable insights that serve as useful points of reference and comparison. A study by Dahodwala et al. [87] in the U.S. Medicare population, estimated annual all-cause costs at $21,960 (€21,000 in 2023) for mild PD and $27,777 (€27,000) for advanced PD, being 48,145$

**Table 2. Average costs (€) per patient in 3 months in Spain in 2013.**

| Hoehn & Yahr Stage | Average Direct Costs | Average Indirect Costs | Average Total Costs |
|---|---|---|---|
| HY I/II | 1.169,26 | 1.790,67 | 1.880,57 |
| HY III | 2.507,29 | 2.798,55 | 4.382,60 |
| HY IV/V | 2.324,51 | 4.221,59 | 6.266,66 |

Table 3. Average costs (in €) in Sweden per H&Y. (EUR/SEK conversion rate in March 2025).

| Sweden Costs in EUR | H&Y I | H&Y II | H&Y III | H&Y IV | H&Y V |
|---|---|---|---|---|---|
| Productivity loss | 3.327,66 | 6.064,47 | 5.701,59 | 3.353,94 | 4.278,15 |
| Informal Care Costs | 159,39 | 674,73 | 2.775,60 | 11.054,61 | 4.987,89 |
| Transport Costs | 40,23 | 109,62 | 103,23 | 185,49 | 96,75 |
| Formal Care Costs | 106,74 | 1.186,47 | 7.133,31 | 27.581,85 | 76.953,42 |
| Drug Costs | 970,47 | 1.646,64 | 2.817,63 | 6.235,47 | 5.419,89 |
| Inpatient and Outpatient Costs | 1.028,25 | 2.041,65 | 2.564,28 | 1.688,85 | 3.333,15 |
| Total | 5.632,74 | 11.723,58 | 21.095,64 | 50.100,21 | 95.069,25 |

(46,000€) the highest decile. Primary PD-related costs were $2,656 (€2,600) per year in mild PD and $6,302 (€6,250) per year in advanced PD. Total PD-related costs (including indirect costs) ranged from $8,751 (€8,700) for mild PD to $14,839 (€14,000) for advanced PD.

As shown, the most significant costs in PD management are medications and formal care. Drug expenses vary significantly, especially for second-line therapies or Device-Assisted Therapies (DAT), such as continuous infusion pumps or DBS, which are particularly costly.

In Spain, the cost of medications such as Rasagiline, Rotigotine, Safinamide, and Opicapone averages €1.500 per year (source: www.vademecum.es). However, as shown in Table 4, these expenses can escalate to more than €40.000 per year due to the use of DAT.

The Formal Care costs refer to home care, nursing services, hospitalisations, and it goes from 7.133€ when the patient has controlled motor fluctuations to 27.581€, for patients considered advanced. The cost of formal care for patients in H&Y V, who are patients who cannot walk and use a wheelchair, is 76.953€. Also, there is an important cost for informal care, which is the cost of the care received by a relative. The burden cost is sometimes underestimated, but severely affects the economy and the QoL of the person who is caring and helping the patient [91,92].

In a longitudinal study in Germany [93], the mean annual cost is 20.095€, and direct costs are 13.158€ for each patient. A total of 2.315€ is considered for hospitalizations, 3.526€ for drugs, and indirect costs accounted for 6.937€ and informal care 2.479€. The costs are stratified into 4 quartile groups based on the UPDRS score. The cost increased for each quartile by 5.000€ to 8.000€. The direct costs of health insurance without drugs (hospitalizations, nursing care, special equipment, rehabilitation) were practically 12.000€.

A study by Winter Y. et al. examined the economic burden of Parkinson's disease across several countries, including Austria, Germany, the Czech Republic, Italy, Portugal, and Russia. Direct costs—comprising inpatient and outpatient care, medical expenses, and rehabilitation—accounted for 60% to 70% of total costs in all countries. Annual per-patient costs ranged from €2,630 to €9,820, increasing with disease severity. In Germany, for example, costs rose from €6,000 per year in mild stages to €16,000 in advanced stages. Similarly, in Austria, the Czech Republic, and Italy, annual costs exceeded €12,000 for advanced patients but remained below €7,000 in earlier stages. The hospitalization cost per day is

Table 4. Average cost of advanced PD therapies per year in a 5-year study.

| | DBS | CSAI | CDLI |
|---|---|---|---|
| Gomez-Inhiesto [88] | 10.643 € | 34.118 € | 41.632 € |
| Vivancos Matellano [89] | 17.895 € | 22.069 € | 46.928 € |
| Valldeoriola [90] | 17.602 € | 28.276 € | 46.797 € |

298€ in Austria, 441€ in Germany, 78,6€ in Italy, and 638€ in Portugal [94]. Rehabilitation is also expensive, being 253€/day in Austria, 172€/day in Germany, 496€/day in Italy, and 268€/day in Portugal. In [95], a study of an Italian cohort was performed, showing that the medication costs are estimated in 24% of direct costs, which range from 1.220 to 1.760€ in a period of 6 months. The costs, however, do not include DAT. Finally, it's noteworthy to discuss costs in the UK based on 2 articles by Findley et al. [96,97]. Both articles examined the economic burden of Parkinson's disease, showing that costs rise steeply with disease progression, especially in advanced stages.

The 2003 study estimated a mean annual cost of nearly £13,500/ €15,800 (2025-adjusted) per patient, increasing six-fold from early (£6,700/ €7,800 at H&Y I) to late-stage (£41,300/ €48,300 at H&Y V) Parkinson's. Around 38% of these costs were borne by the NHS, with the rest split between social services and personal/family expenses.

The 2011 study focused on advanced Parkinson's and linked costs to the percentage of the day patients spend in the OFF state (when medication wears off). Annual costs ranged from £41,000/ €48,000 to over £99,400/ €116,300 as OFF time increased from <25% to >75% of the day. In this context, 93% of costs were related to care, with only 7% to direct medical services.

Together, these studies reveal that the main cost driver in Parkinson's is care needs, not drugs or hospital visits, especially in the advanced stages of the disease.

## 5. Cost-benefit scenarios using a medical device

The objective of this study is to examine the potential impact of STAT-ON™ on reducing hospitalisation costs, productivity loss, and home assistance expenses by improving QoL and disease management when introduced in clinical praxis in the considered countries.

The rationale behind this proposal is that STAT-ON™ enhances healthcare services by enabling a more precise and objective evaluation of patients. This improved assessment allows clinicians to prescribe more appropriate treatments, which, in turn, improves patients' quality of life and effectively reduces symptom severity.

Reducing symptoms severity in Parkinson's Disease (PD) has a direct impact on healthcare costs. As the disease progresses, treatment becomes increasingly expensive. In the early stages, patients typically rely on levodopa-based medication, but as motor fluctuations emerge, dopaminergic inhibitors are introduced to extend levodopa's effects. When fluctuations become difficult to manage, leading to impaired gait, postural instability, FoG, or dyskinesia, more advanced treatments like DATs are considered. These therapies, while effective, are significantly more costly than oral medication.

However, reducing OFF periods, increasing ON time, and minimizing FoG episodes can decrease fall risk, leading to improved QoL and lower healthcare costs by reducing hospitalisations, outpatient visits, medical consultations, and nursing services. While indirect costs such as productivity loss and caregiver burden do not directly impact healthcare budgets, they contribute significantly to the overall economic and social burden of the disease. Effective disease management through continuous monitoring and timely interventions may help mitigate these costs.

Conducting a cost-benefit analysis of STAT-ON™ presents challenges due to the lack of standardized healthcare cost assessment methods in PD. Additionally, differences in healthcare systems, labour costs, and treatment availability across Europe complicate cost estimations and comparisons. These variations highlight the need for country-specific analyses when evaluating economic feasibility.

To address this, we conducted a cost-benefit estimation of STAT-ON™ based on a literature review across Spain, Italy, Germany, the UK, and Sweden. By analysing studies on PD monitoring technologies and healthcare costs, we developed country-specific projections to assess their financial impact. This approach provides insight into how continuous symptom monitoring could influence healthcare expenditures, improve patient outcomes, and optimize resource allocation.

While these estimates have inherent limitations, they offer a useful starting point for evaluating the economic feasibility of integrating STAT-ON™ into clinical practice. Future large-scale studies will be essential to validate these findings, refine cost estimates, and further explore the long-term benefits of STAT-ON™ in PD management.

We identify the following key points as important considerations across different European countries:

• Particularities of the National Health Systems in each country

• Different cost of drugs and other treatments

• Inflation (as many studies found came from the early 2000s).

However, some assumptions can be made and will be presented in this section, to extract some conclusions at this respect.

1. The yearly cost of a STAT-ON™ in Europe/US is around 2500€±500€. Thus, assuming that the health professional recommends to the patient the use of STAT-ON™ at least twice every year, and that a single STAT-ON™ is on average used 30 times every year (the year has 52 weeks and STAT-ON™ is usually used in periods of 7 days), it is possible to estimate a yearly cost of the device per patient in 166€/patient.

2. The second assumption we take for this cost-benefit analysis is the improvement shown by patients when a correct medication is introduced in their therapy. According to [51], the reduction of OFF hours per day when applying Safinamide, Entacapone, Opicapone, or Rasagiline to PD patients with fluctuations is similar, ranging from 1,1 hours for Entacapone to 2 hours for Opicapone, being an average of 1,37 hours. We thus, assume that these patients could be considered from a moderate state to a mild state at the moment that they start with these dopaminergic inhibitors. On the other hand, the reduction of OFF hours per day with an CDLI therapy is assumed to be around 4,5 hours and 2,5 hours for CSAI. In this case, we can assume that the patient could be considered a moderate patient instead of an advanced one in terms of OFF hours per day.

3. The third assumption is that the use of STAT-ON™ may enable neurologists to detect motor fluctuations or advanced symptoms earlier, potentially leading to the earlier initiation of optimized therapies. However, this could also result in a significant increase in drug-related costs, particularly due to greater use of dopamine agonists or device-aided therapies. In this regard, not all advanced treatments have demonstrated clear cost-effectiveness. While DBS has been shown to be cost-efficient [98], the cost-benefit profiles of CDLI and CSAI remain uncertain due to their high associated costs [99]. However, the reduction of OFF hours and improvement in QoL is significant in all the therapies, which eventually motivates the Health Systems to fund these therapies [100]. On the other hand, Opicapone [101], Safinamide [102], Rasagiline [103], Entacapone [104], and Rotigotine [105], among others, have demonstrated to be cost-effective for health systems, which means that the use of drugs complementing levodopa-based medication does not increase the costs of the Health Systems. In the article by Heald et al. [105], due to the use of Safinamide and Apomorphine, there is a reduction of 28% in hospitalizations.

4. The final assumption is that we will consider 3 stages in PD. This distinction is based on the number of different considerations, evaluations, scales, and the need to reduce data for the analysis. The 3 stages of PD are mild, moderate, and advanced.

In this cost-benefit analysis, we examine two scenarios. The first one (depicted in Fig 3) shows a patient who exhibits moderate symptoms of Parkinson's disease. However, despite having moderate symptoms, the patient remains diagnosed as a patient with mild symptoms, a condition that affects at least 20% of PD patients [41]. Although the patient has received a prescription with medication for mild symptoms, the direct costs of caring for the patient are moderate. The rationale for this study is to demonstrate that the introduction of STAT-ON™ technology can accurately detect these underdiagnosed moderate symptoms, allowing for optimisation of the medication regime. Although the more effective medication may be costlier, it can reduce direct expenses by alleviating PD symptoms and improving the patient's quality of life.

The second scenario (illustrated in Fig 4) depicts the case of patients who are in an advanced Parkinson's disease stage. Despite having severe symptoms, the patient is not diagnosed as an advanced stage (as explained in Section

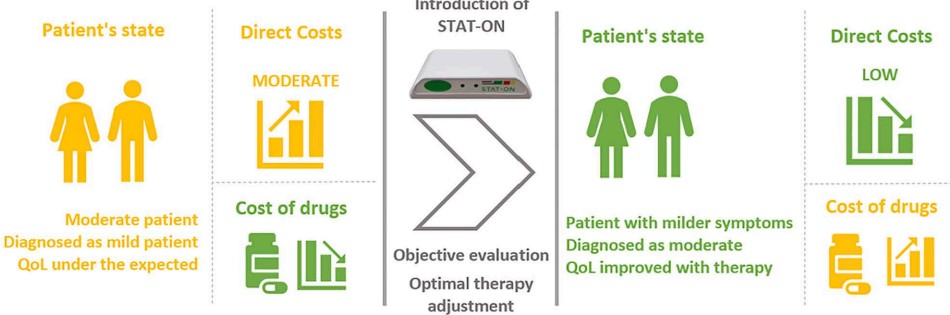

**Fig 3. Scenario 1. Moderate patient with STAT-ON™.**

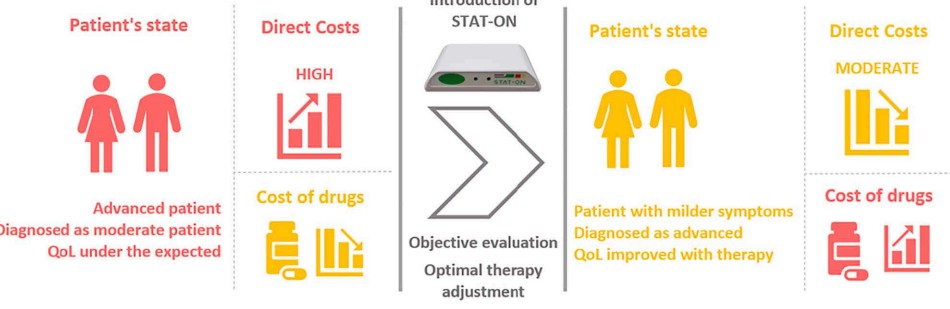

**Fig 4. Scenario 2. Advanced patient with STAT-ON™.**

2.2), which directly impacts negatively on their quality of life. In this case, the direct costs of caring for the patient are high, reflecting the severity of their condition. The introduction of STAT-ON™ technology can improve the diagnosis, thus enhancing, the patient's quality of life, resulting in a reduction in direct costs but, in contrast, increasing the cost of medication. By accurately detecting the underdiagnosed advanced symptoms, STAT-ON™ can optimise the medication regime, resulting in better symptom control and improved patient outcomes. In summary, although the more effective medication may be initially expensive, it can lower direct costs by reducing hospitalisation and other healthcare expenses. The use of STAT-ON™ might result in better clinical outcomes and lower overall costs of care for patients with advanced underdiagnosed Parkinson's disease.

Below, we propose a model considering the introduction of STAT-ON™ in the healthcare system. The information extracted for the estimation of costs (Table 5) has been extracted from [84,85] for Spain, [94,95] for Italy, [86] for Sweden, [83,93,94] for Germany, and [96,97] for UK (in the referred papers, costs appear in pounds and the euro equivalence has been applied). The estimated costs for Direct costs (first column) are considered, excluding the medication cost, which is presented in the second column. The number of PD patients diagnosed according to [82] and stratified by the stage of the disease are shown in the third column (milder, moderate and advanced). The percentages, which have been averaged from the data of this document (Table 1), are 45% for mild patients, 35% for moderate patients, and 20% for advanced patients.

Table 6 shows an estimation of the number of mild PD patients with motor fluctuations (MF) not correctly diagnosed and the number of moderate patients not diagnosed as APD in five different countries. The number of patients has been

**Table 5. Average medication costs in five European regions (column 1), average costs of medication (column 2) and number of patients in Spain, Sweden, Germany, Italy and the UK (column 3).**

|  | Column1: Direct costs in Hospitals in € (no drugs) |  | Column 2: Cost of medication (€) |  | Column 3: N. patients |  |
|---|---|---|---|---|---|---|
| Milder patients | Spain | 5.600 | Spain | 1.904 | Spain | 47.914 |
|  | Sweden | 2.180 | Sweden | 1.308 | Sweden | 9.000 |
|  | Germany | 2.000 | Germany | 1.500 | Germany | 172.350 |
|  | Italy | 6.460 | Italy | 2.040 | Italy | 54.000 |
|  | UK | 6.039 | UK | 1.907 | UK | 46.776 |
| Moderate patients | Spain | 8.680 | Spain | 2.951 | Spain | 37.266 |
|  | Sweden | 7.133 | Sweden | 2.817 | Sweden | 7.000 |
|  | Germany | 9.105 | Germany | 5.000 | Germany | 134.050 |
|  | Italy | 8.677 | Italy | 2.740 | Italy | 42.000 |
|  | UK | 12.376 | UK | 3.908 | UK | 36.381 |
| Advanced patients | Spain | 10.800 | Spain | 3.672 | Spain | 21.295 |
|  | Sweden | 52.300 | Sweden | 5.830 | Sweden | 4.000 |
|  | Germany | 54.006 | Germany | 15.976 | Germany | 76.600 |
|  | Italy | 9.943 | Italy | 3.140 | Italy | 24.000 |
|  | UK | 28.522 | UK | 9.007 | UK | 20.789 |

**Table 6. Two rows presenting the number of patients not diagnosed with MF and as APD. The different columns present the estimation of non-correctly diagnosed patients per country, the number of these patients potentially detectable by STAT-ON™ and the number of these patients using STAT-ON™ when considering a market penetration of 20%.**

|  | Country | Estimation of non-diagnosed patients | Estimation of the n. of patients with possibilities to be detected with STAT-ON™ (90%) | N. of patients detected by STAT-ON™ with a market penetration of 20% |
|---|---|---|---|---|
| Number of patients not diagnosed with MF | Spain | 12.150 | 10.935 | 2.187 |
|  | Sweden | 1.620 | 1.458 | 292 |
|  | Germany | 14.580 | 13.122 | 2.624 |
|  | Italy | 9.720 | 8.748 | 1.750 |
|  | UK | 10.287 | 9.258 | 1.852 |
| Number of patients not diagnosed as APD | Spain | 18.900 | 17.010 | 3.402 |
|  | Sweden | 2.520 | 2.268 | 454 |
|  | Germany | 22.680 | 20.412 | 4.082 |
|  | Italy | 15.120 | 13.608 | 2.722 |
|  | UK | 16.002 | 14.402 | 2.880 |

estimated based on the DEEP study [41], Observe-PD study [44], DISCREPA study [42], and PARADISE study [43] (the applied estimation is the 18% for the considered mild patients and 36% for the considered moderate patients). According to these estimations, the total number of PD patients is approximately 150,000 in Spain, 20,000 in Sweden, 180,000 in Germany, 120,000 in Italy, and 127,000 in the UK.

Also, Table 6 shows the estimation of the total number of underdiagnosed PD patients with possibilities to be correctly detected when using STAT-ON™ (this is according to the three published clinical studies performed with 75, 41, 15, and 23 PD patients, respectively [19,20,22,106], where STAT-ON™ has demonstrated a 0,9 sensitivity and specificity for the correct detection of motor fluctuations). Thus, in the corresponding column it is shown a 90% of the PD patients that, thanks to STAT-ON™, would be correctly identified.

Finally, in the last column of Table 6, we estimate a penetration market of 20%. In this case, we only contemplate using STAT-ON™ in 20% of hospitals in different countries due to adoption barriers. We understand that the market penetration will not be 100% and that the technology will be introduced gradually. As a conservative number, we consider the use of STAT-ON™ only in 20% of Movement Disorders Units, Hospitals, General neurologists, and private clinics, and this percentage will increase over time when the usefulness of STAT-ON™ becomes evident.

Finally, considering the presented numbers in Table 6, all the aforementioned assumptions, and both scenarios sketched in Figs 3 and 4, we are able to make an estimation of the savings when using STAT-ON™. To estimate these potential savings, we can make the following assumptions and then proceed with a calculation using the following equations 1 and 2:

- We will assume that Parkinson's Disease patients correctly identified, monitored, and treated with the use of STAT-ON™ would be considered as if their condition were better than it is (milder instead of moderate, or moderate instead of APD), at least in a high percentage of cases (we will assume that it is true in approximately 80% of cases).

- The cost of medications (treatment) is higher in a moderate state than in a milder state, and in APD than in moderate (this factor works against the savings). When the patient is treated with optimal medication, medication costs (MC) will be more ex-pensive, but the direct costs (DC) will be reduced given the improvement in the PD patient's state.

- It is necessary to consider the cost of using STAT-ON™ (it has been previously stated as 166€/patient per year).

- The DC and MC have been stratified by PD stages (Mild, Moderate, Advanced) and are quantified according to Table 5.

The global hypothesis is that the savings will come from a correct diagnosis of the patients and their correct and adapted treatment.

Equation 1 shows the calculation of the possible savings according to the Scenario 1 (Fig 3) and the above-presented hypothesis.

$$Saving = \ [(DC_{Mod} - DC_{Mild}) * 0,8 - (MC_{Mod} - MC_{Mild}) - STAT-ON_{cost})] * N.pat. \tag{1}$$

Similarly, Equation 2 shows the calculation for Scenario 2 (Fig 4).

$$Saving = [(DC_{Adv} - DC_{Mod}) * 0,8 - (MC_{Adv} - MC_{Mod}) - STAT-ON_{cost})] * N.pat. \tag{2}$$

where DC states for Direct Costs (advanced, moderate, or mild patients), MC for Medication Costs (Advanced, Moderate, or Mild patients), and N.Patients are the patients considered in the last column of Table 6.

Table 7 presents the total savings applying Equations 1 and 2. The left part of the Table presents the savings considering the use of STAT-ON™ for detecting MF in those patients that are not well evaluated, considering the reduction in Direct Costs for Care and Hospitalization, but adding the extra costs in Medication for Moderate patients and the use of STAT-ON™. The right part of the table is the same, but for the correct detection and evaluation of patients with advanced symptoms.

**Table 7. Total savings in EUR in Health Systems in 5 countries considering the use of STAT-ON™ for the correct detection of symptoms.**

| Savings with Moderate patients in € | | Savings with Advanced patients in € | |
|---|---|---|---|
| Spain | 4.082.692 | Spain | 4.195.346 |
| Sweden | 955.865 | Sweden | 19.045.757 |
| Germany | 9.025.312 | Germany | 137.817.742 |
| Italy | 2.363.126 | Italy | 1.906.934 |
| UK | 7.720.983 | UK | 31.342.029 |

## 6. Discussion

The analysis of costs and benefits related to the adoption of a wearable device is undeniably intricate and demands careful consideration of multiple variables. In medical settings, there exists a recognised need for employing objective instruments that can identify symptoms accurately in domestic settings to ensure appropriate treatment recommendations for patients [107]. Parkinson's Disease is a highly complex and diverse condition, and patients receive multiple therapies based on the manifestation of motor or non-motor symptoms and the progression of the disease. To deal with this complexity, in this paper, PD patients have been classified into three distinct and reasonable groups: those who have recently been diagnosed and are being treated with levodopa, those who are in the moderate stages of the disease and experience motor fluctuations, necessitating the use of dopaminergic inhibitors to alleviate these symptoms, and those who are deemed advanced PD patients.

The methodology employed in this study is underpinned by existing literature on pharmacoeconomic studies, drug cost-benefit analyses, and the validation of STAT-ON™, which allows for a more comprehensive assessment of PD patients across these three groups.

All commercial therapies have been subject to positive cost-benefit studies, and when combined with the cost of implementing STAT-ON™, the introduction of the device into Health Systems becomes an affordable solution that addresses the issue of accurate clinical evaluations. The resulting cost savings, although they are very complicated and it is only possible to make a reasonable approximation, they are undeniably beneficial. It should be noted that this process not only generates cost savings but also significantly improves the patient's quality of life.

One interesting situation occurs in Italy, where the detection of moderate patients using STAT-ON™ results in greater cost savings than the detection of patients with advanced symptoms, likely due to the high cost of DAT. However, in Sweden, the UK, and Germany, detecting patients with advanced symptoms using STAT-ON™ leads to substantial savings, likely due to more frequent hospitalisations and institutionalisation for comprehensive evaluations.

The primary aim of STAT-ON™ is to reduce hospitalisations and direct costs for care by improving the patient's quality of life through the prescription of appropriate therapies based on a more comprehensive and objective evaluation. Additionally, STAT-ON™ can serve as a telemedicine (TM) tool for remote monitoring of PD patients [108], which drastically reduces the need for doctor's visits. In this study, it is demonstrated significant improvements in daily living activities, depression, apathy, freezing of gait, balance, and frailty are demonstrated in the TM group. The STAT-ON™ wearable sensor provided objective motor assessments, detecting gait alterations and freezing episodes with greater sensitivity than patient-reported measures. Cost-effectiveness analyses indicated incremental cost-effectiveness ratios (ICERs) ranging from €91.55 for non-motor symptoms to €1,677.4 for frailty, suggesting that TM can serve as an efficient complementary approach. While no significant differences in quality-adjusted life years (QALYs) were observed—likely due to the study's short duration (8 months)—clear clinical benefits were evident. Additionally, the TM group exhibited increased physical activity and greater awareness of fall risk, without an associated rise in fall-related incidents [109]. In another study presented in Spain, the costs of assessing patients with a platform remotely and using STAT-ON™ were reduced from €5.108,26 to €2.243,07 per patient. The quality of life and other indicators from other questionnaires also improved [110].

Beyond clinical care, STAT-ON may also support research planning. Objective data on motor fluctuations and daily activity can help design more homogeneous study groups, tailor exercise interventions, and provide reliable and objective follow-up data. By enabling closer, personalized monitoring, the device may also reduce participant drop-out and improve adherence, strengthening the quality of clinical trials.

Despite the increasing integration of technology in healthcare, adoption barriers and scepticism persist, particularly regarding certain monitoring devices, such as wrist-worn sensors for assessing axial symptoms in Parkinson's Disease (PD). A key limitation of wrist-worn devices is their susceptibility to random upper limb movements, which can hinder accurate detection of motor complications affecting the trunk, lower limbs, and neck. In contrast, STAT-ON™ offers

an alternative approach by leveraging its waist-worn placement, enabling more precise monitoring of axial symptoms with a single device. Its machine learning algorithms were developed through a clinician-supervised process, utilising video-recorded assessments in home environments during both ON and OFF states. This resulted in the largest inertial database of PD patients under these conditions (92 participants). Furthermore, the algorithm's clinical validation was conducted using standardised methods, including UPDRS scoring, Hauser diaries, and follow-up calls for verification, ensuring rigorous and reliable motor symptom assessment [20–22,106].

These findings suggest that STAT-ON™ provides a highly accurate detection of multiple motor symptoms in Parkinson's Disease. It might also indirectly contribute to overall health (systemic comorbidities, diabetes or cardiovascular diseases) by improving mobility, reducing complications such as falls, and enabling better adaptation of therapy. Its ability to provide objective assessments may support more precise therapeutic adjustments, potentially improving patients' quality of life while reducing healthcare costs associated with hospitalizations, medical consultations, and disease management.

## 7. Limitations of the study

While this study presents a cost-benefit analysis supporting the integration of the STAT-ON™ wearable device in Parkinson's Disease (PD) management, certain limitations must be acknowledged. First, the analysis relies on economic data from various European healthcare systems, which differ in their structure, reimbursement policies, and cost of medical care, potentially affecting the generalizability of the findings. Second, the study assumes that early detection and optimised treatment lead to improved QoL and cost savings, but long-term clinical studies confirming these benefits are still needed. Third, while the model accounts for direct medical costs, indirect costs such as caregiver burden and lost productivity were not fully incorporated, which could impact the overall cost-effectiveness of the device. Additionally, market penetration of STAT-ON™ was estimated conservatively at 20%, but real-world adoption rates could vary due to factors such as clinician acceptance, patient adherence, and regulatory hurdles. Finally, the accuracy of motor fluctuation detection was based on previous validation studies, but further large-scale trials in diverse clinical settings would be beneficial to confirm its effectiveness across different patient populations. Future research should aim to address these limitations by conducting longitudinal studies that assess real-world outcomes, healthcare utilisation, and patient-reported benefits over extended periods.

## 8. Conclusions

STAT-ON™ is a wearable medical device with a strong scientific foundation, supported by over 80 scientific publications, and partially recommended by the National Institute for Health and Care Excellence (NICE). It addresses a critical gap in the objective evaluation of Parkinson's Disease patients, providing clinicians with reliable data to optimise treatment strategies. Accurate patient assessment enables more precise therapy adjustments, ultimately improving QoL.

This wearable sensor has proven useful in several ways: it supports physicians in optimizing treatment plans, assists in identifying patients who are suitable for second-line therapies—such as deep brain stimulation (DBS), continuous dopaminergic infusions (like apomorphine or levodopa-carbidopa intestinal gel), or other advanced treatments typically considered when first-line medications no longer provide adequate symptom control—enhances the fine-tuning of infusion dosages, increases patient awareness of their symptoms, supplements or even replaces traditional patient diaries, and empowers patients to better recognize and understand their own Parkinson's-related symptoms.

An increase in QoL is associated with a reduction in direct healthcare costs, such as hospitalisations and medical visits. The reduction in hospitalisations is due to the reduction in falls, or complications related to frailty caused by Parkinson's. Also, the integration of specialized nursing services into movement disorders units is increasingly common, and training these professionals in STAT-ON use could further enhance efficiency.

However, it may also lead to higher medication expenses, as more advanced disease stages require increased pharmacological intervention. Dopaminergic inhibitors such as MAO-B and COMT inhibitors are more expensive than oral

levodopa, and DATs represent an even greater financial burden. While the inclusion of STAT-ON™ introduces an additional cost, the QoL improvements and reductions in direct healthcare expenses help offset these costs, as illustrated in Table 7.

Beyond its direct impact on patient management, STAT-ON™ is a feasible tool for routine clinical practice, contributing to long-term healthcare cost reductions. Given the rising costs associated with an aging population and increased life expectancy in Europe, Asia, and North America, efficient healthcare solutions are critical to mitigating future economic burdens.

It is important to note that this study does not account for the potential positive impact of STAT-ON™ on reducing indirect costs related to productivity loss for patients and caregivers. These costs, borne by public administrations and society, can be also significant. However, due to substantial variations in healthcare systems, social support structures, and economic conditions across countries, indirect costs were not included in this analysis. A country-specific approach would be required to assess the impact of productivity loss, caregiver burden, and healthcare expenses in different regions.

In conclusion, while this study demonstrates the feasibility and cost-benefit potential of STAT-ON™ in terms of QoL improvement and direct cost reduction, further research is needed to fully assess its economic impact, particularly regarding indirect costs. A more comprehensive analysis, incorporating country-specific data, would provide a clearer picture of STAT-ON™'s overall cost-effectiveness in PD management.

## Supporting information

**S1 Table. Cost table.** Table with the data used for the cost benefit study.
(XLSX)

## Author contributions

**Conceptualization:** Andreu Català, Daniel Rodriguez-Martin, Joan Cabestany.

**Data curation:** Daniel Rodriguez-Martin.

**Formal analysis:** Andreu Català.

**Investigation:** Daniel Rodriguez-Martin.

**Methodology:** Joan Cabestany.

**Software:** Daniel Rodriguez-Martin.

**Supervision:** Andreu Català, Joan Cabestany.

**Validation:** Joan Cabestany.

**Visualization:** Daniel Rodriguez-Martin.

**Writing – original draft:** Daniel Rodriguez-Martin.

**Writing – review & editing:** Andreu Català, Joan Cabestany.

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
