## [Decision Letter · Decision Letter 0]

25 Jul 2025

Dear Dr. Català,

We look forward to receiving your revised manuscript.

Kind regards,

Joel Montané, PhD

Academic Editor

PLOS ONE

2. Please note that PLOS One has specific guidelines on code sharing for submissions in which author-generated code underpins the findings in the manuscript. In these cases, all author-generated code must be made available without restrictions upon publication of the work. Please review our guidelines at https://journals.plos.org/plosone/s/materials-and-software-sharing#loc-sharing-code and ensure that your code is shared in a way that follows best practice and facilitates reproducibility and reuse.

Additional Editor Comments (if provided):

Reviewers' comments:

Reviewer's Responses to Questions

**Comments to the Author**

1. Is the manuscript technically sound, and do the data support the conclusions?

Reviewer #1: Yes

Reviewer #2: Yes

2. Has the statistical analysis been performed appropriately and rigorously?

Reviewer #1: Yes

Reviewer #2: Yes

3. Have the authors made all data underlying the findings in their manuscript fully available?

Reviewer #1: Yes

Reviewer #2: Yes

4. Is the manuscript presented in an intelligible fashion and written in standard English?

Reviewer #1: Yes

Reviewer #2: Yes

Reviewer #1: I think that for a better reading of the image, fig.1 should be changed. As the reading system is from left to right, I suggest to simply swap “Doctor’s visit” and “STAT-ON monitoring” for “Improving QOL” and “Therapy adjustment or change”.

Line 139) If I understood the sentence right, I think it is possible to delete the “in” at the end and just leave the citation.

From line 133 to 140) Since the section is about the “summary of STAT-ON validation,” I think this part fits better in the conclusion section. Maybe removing it from here could contribute to keep the text concise.

From line 171 to 214) despite the really interesting section, I’m afraid it could appear a bit out of context. In my opinion, it is better to keep the text concise and focused on the main topic. Maybe the last part of the section (line 212, 214) could be used in the introduction to better emphasize the need for a reliable and objective system to monitor patients’ conditions in order to adjust the treatment.

Your study appears to be very meticulous and detailed. I think that section 5 alone could be enough for your article. It is certainly important to know all the background information in order to understand how the device can be used to reduce government spending. However, such a lengthy text may distract from the purpose of your work. I recommend reducing the sections that address topics too far removed from the main topic.

Since you investigate the utilization of devices in monitoring patient’s condition, if I may, I suggest you to consider these interesting papers about the topic.

-Libero, T. D., Carissimo, C., Cerro, G., Abbatecola, A. M., Marino, A., Miele, G., Ferrigno, L., & Rodio, A. (2023). Motor abilities analysis using a standardized tapping test enhanced by a detailed processing stage: gender and age comparison. 2023 IEEE International Symposium on Medical Measurements and Applications (MeMeA), 1–6. https://doi.org/10.1109/memea57477.2023.10171922

-Krokidis, M. G., Dimitrakopoulos, G. N., Vrahatis, A. G., Tzouvelekis, C., Drakoulis, D., Papavassileiou, F., Exarchos, T. P., & Vlamos, P. (2022). A Sensor-Based Perspective in Early-Stage Parkinson’s Disease: Current State and the Need for Machine Learning Processes. Sensors, 22(2), 409. https://doi.org/10.3390/s22020409

Reviewer #2: General Comments

The manuscript addresses an important topic related to the use of portable monitoring (STAT-ON device) in the management of Parkinson’s disease, with potential implications in terms of cost-effectiveness and clinical practice. The subject is highly relevant given the increasing prevalence of Parkinson’s disease and the growing need for optimized multidisciplinary care.

Overall, the article is well-structured and informative. However, I believe the authors could further strengthen the manuscript by introducing improvements that enhance its clarity, justification, and coherence, as well as by expanding its scientific robustness.

Specific Comments

Demographics and Age Factor – For the first paragraph of the Introduction:

- It would be advisable to cite the prevalence of Parkinson’s disease in Europe.

- Regarding age as a risk factor: I suggest emphasizing, elaborating, and referencing that advanced age is one of the primary risk factors for the development of Parkinson’s disease, as both its incidence and prevalence significantly increase with population aging. Including this information would reinforce the epidemiological justification of the study.

- Concerning demographic projections: it would be appropriate to include a citation of projected growth in the elderly population toward 2035 or 2050. This demographic trend is expected to increase the burden of Parkinson’s disease, thereby reinforcing the public health relevance of the problem.

Cost-Benefit Framework:

- I recommend reinforcing the discussion around how continuous monitoring contributes to more efficient patient management, particularly by addressing comorbidities, for example, by reducing the risk of falls, hospitalizations, or complications related to frailty.

- I also suggest clarifying that the currently available evidence regarding the STAT-ON device primarily supports its use in the management of direct comorbidities (motor symptoms, falls, freezing of gait). It would be helpful to explicitly distinguish these direct comorbidities from indirect or systemic conditions (such as diabetes or cardiovascular diseases), in which the device does not act directly but could potentially contribute indirectly by improving the patient’s quality of life and holistic care.

Evidence and References – Regarding the paragraph between lines 132–140:

- I recommend including specific references to support the claims made regarding its clinical use.

- Although the paragraph is informative, its clarity could be improved by breaking up long sentences, avoiding extensive lists within a single sentence, and using clearer logical connectors.

Clinical Integration and Optimization – Suggested Comment for Lines 152–158:

- This paragraph highlights that the challenges mentioned in clinical assessments represent a significant opportunity to optimize time management during follow-up visits, which is an notable strength in improving both clinical efficiency and effectiveness. In this regard, the incorporation of specialized nursing services into movement disorders units is becoming increasingly common. Therefore, providing these professionals with training in the use of the STAT-ON device could further enhance this benefit. For example, the Hospital Vall d’Hebron has recently implemented this type of support, allowing for more effective and efficient patient care through more comprehensive pre-evaluations and better-organized neurological consultations.

From Line 171 Onward – Section 2.3: Improving Quality of Life with Parkinson’s Disease Therapies

- It may be useful to begin by clearly stating that there are two main types of therapies: pharmacological and non-pharmacological. This would help structure the section more clearly for the reader and emphasize that both are fundamental components of comprehensive Parkinson’s disease management.

- Additionally, I would suggest that the authors include a more explicit mention of non-pharmacological options (such as physiotherapy, structured exercise, cognitive stimulation, or psychological support), which are also essential in quality-of-life-oriented care for Parkinson’s disease.

- It would also be advisable to emphasize that therapeutic decisions must always be individualized, taking into account the patient's clinical characteristics, preferences, and resource availability, thereby reinforcing the patient-centered approach.

- Finally, I believe it would be highly relevant to include (ideally toward the end of the paragraph) how monitoring tools such as STAT-ON could benefit both types of therapy. While its potential has already been mentioned in relation to pharmacological treatments, it could also provide data on mobility and daily activity that may help tailor rehabilitation programs or prescribe physical exercise. Including this point could enrich the manuscript and underscore the value of objective monitoring in clinical decision-making for both pharmacological and non-pharmacological treatments.

Research Planning and Drop-Out Reduction:

- I would suggest incorporating an important point in the manuscript regarding the potential utility of the STAT-ON device in research planning and the reduction of participant drop-out rates.

- The authors could consider the possibility that data collected in advance by the device may help optimize the planning of various clinical trials, for instance, in studies focused on physical exercise. Objective data on motor fluctuations and daily activity could contribute to better structuring of training microcycles tailored to each patient’s profile, and also to the homogenization of intervention and control groups, an essential factor in clinical study design.

- Furthermore, it would be worth highlighting that the use of objective monitoring with STAT-ON during participant follow-up could yield more reliable and detailed data, thereby enhancing the quality of the resulting evidence. This tool could also help reduce drop-out rates by enabling more personalized adjustments and closer monitoring, thus improving participant adherence to research protocols.

Broader Application to Other Therapies:

- In line with the previous comment related to research, the manuscript could also consider how STAT-ON might assist in objectifying outcomes in more “subjective” therapies such as manual therapy, osteopathy, physiotherapy, and other alternative interventions.

- Having access to quantitative data on motor parameters or daily activity could provide stronger evidence of therapeutic effects and facilitate the integration of such interventions into clinical practice guided by objective outcome measures.

Final Comment

I believe the manuscript would significantly benefit from the suggested clarifications and expansions, which would help strengthen its scientific rigo. I encourage the authors to incorporate these aspects to increase the overall impact of their work.

**Do you want your identity to be public for this peer review?** For information about this choice, including consent withdrawal, please see our Privacy Policy

Reviewer #1: No

Reviewer #2: No

---

## [Author Response · Author response to Decision Letter 1]

24 Sep 2025

Response to Reviewer #1

Thank you for your review and your useful comments for improving this paper. Following the editor's recommendations, we have prepared a revised version of our manuscript addressing your concerns.

Let us briefly indicate the modifications concerning the comments.

Comments

I think that for a better reading of the image, fig.1 should be changed. As the reading system is from left to right, I suggest to simply swap "Doctor's visit" and "STAT-ON monitoring" for "Improving QOL" and "Therapy adjustment or change".

Figure 1 has been modified to follow a left-to-right reading direction, placing “Doctor’s visit” and “STAT-ON monitoring” before “Therapy adjustment or change” and “Improving QoL” for greater clarity.

Line 139) If I understood the sentence right, I think it is possible to delete the "in" at the end and just leave the citation.

Yes, the suggestion is correct and we have modified it.

From line 133 to 140) Since the section is about the "summary of STAT-ON validation," I think this part fits better in the conclusion section. Maybe removing it from here could contribute to keep the text concise.

The proposed text has been removed from this section and incorporated into the Conclusions to improve conciseness and coherence.

From line 171 to 214) despite the really interesting section, I'm afraid it could appear a bit out of context. In my opinion, it is better to keep the text concise and focused on the main topic. Maybe the last part of the section (line 212, 214) could be used in the introduction to better emphasize the need for a reliable and objective system to monitor patients' conditions in order to adjust the treatment.

We have revised the introduction of this section to clarify that STAT-ON alone cannot directly improve QoL, but by guiding appropriate therapy, it enables interventions that do. We now explicitly state that the section reviews the main treatments and their proven impact on QoL. This way, we focus the reader on the topic of the article.

Your study appears to be very meticulous and detailed. I think that section 5 alone could be enough for your article. It is certainly important to know all the background information in order to understand how the device can be used to reduce government spending. However, such a lengthy text may distract from the purpose of your work. I recommend reducing the sections that address topics too far removed from the main topic.

We have reduced some sections, especially Section 2, and parts from Sections 1 and 3. The background is essential for understanding Section 5. We have eliminated all the explanation of the validation of STAT-ON but we kept the references and the main algorithm validation. We have tried to reduce some parts, but added others due to others comments. We have

tried to deal with all the comments.

Since you investigate the utilization of devices in monitoring patient's condition, if I may, I suggest you to consider these interesting papers about the topic.

-Libero, T. D., Carissimo, C., Cerro, G., Abbatecola, A. M., Marino, A., Miele, G., Ferrigno, L., & Rodio, A. (2023). Motor abilities analysis using a standardized tapping test enhanced by a detailed processing stage: gender and age comparison. 2023 IEEE International Symposium on Medical Measurements and Applications (MeMeA), 1–

6. https://doi.org/10.1109/memea57477.2023.10171922

-Krokidis, M. G., Dimitrakopoulos, G. N., Vrahatis, A. G., Tzouvelekis, C., Drakoulis, D., Papavassileiou, F., Exarchos, T. P., & Vlamos, P. (2022). A Sensor-Based Perspective in Early- Stage Parkinson's Disease: Current State and the Need for Machine Learning Processes.

Sensors, 22(2), 409. https://doi.org/10.3390/s22020409

We have added these references in the paper (Section 1) as the reviewer suggest. We think they improve the justification and the use of wearables in the objective monitoring of Parkinson’s Disease.

Response to Reviewer #2

Thank you for your review and your useful comments for improving this paper. Following the editor's recommendations, we have prepared a revised version of our manuscript addressing your concerns.

Let us briefly indicate the modifications concerning the comments.

Comments

General Comments

The manuscript addresses an important topic related to the use of portable monitoring (STAT- ON device) in the management of Parkinson's disease, with potential implications in terms of cost-effectiveness and clinical practice. The subject is highly relevant given the increasing prevalence of Parkinson's disease and the growing need for optimized multidisciplinary care. Overall, the article is well-structured and informative. However, I believe the authors could further strengthen the manuscript by introducing improvements that enhance its clarity, justification, and coherence, as well as by expanding its scientific robustness.

We sincerely thank the reviewer for the positive overall evaluation of our work and for recognizing its relevance. We also appreciate the constructive suggestions provided, which have helped us improve the clarity, coherence, and scientific robustness of the manuscript.

Specific Comments

Demographics and Age Factor – For the first paragraph of the Introduction:

- It would be advisable to cite the prevalence of Parkinson's disease in Europe.

- Regarding age as a risk factor: I suggest emphasizing, elaborating, and referencing that advanced age is one of the primary risk factors for the development of Parkinson's disease, as both its incidence and prevalence significantly increase with population aging. Including this information would reinforce the epidemiological justification of the study.

- Concerning demographic projections: it would be appropriate to include a citation of projected growth in the elderly population toward 2035 or 2050. This demographic trend is expected to increase the burden of Parkinson's disease, thereby reinforcing the public health relevance of the problem.

We have added data on projections for 2040 and 2050 based on two published studies, highlighting how aging and increased life expectancy are driving the sharp rise in patient numbers. In addition, we have included prevalence figures for Europe.

Cost-Benefit Framework:

- I recommend reinforcing the discussion around how continuous monitoring contributes to more efficient patient management, particularly by addressing comorbidities, for example, by reducing the risk of falls, hospitalizations, or complications related to frailty.

In the final section of Discussions, we emphasize the role of STAT-ON by highlighting its advantages and explaining how it can help reduce falls, hospitalizations, and complications related to frailty.

- I also suggest clarifying that the currently available evidence regarding the STAT-ON device primarily supports its use in the management of direct comorbidities (motor symptoms, falls,

freezing of gait). It would be helpful to explicitly distinguish these direct comorbidities from indirect or systemic conditions (such as diabetes or cardiovascular diseases), in which the device does not act directly but could potentially contribute indirectly by improving the patient's quality of life and holistic care.

We have added a short paragraph in Discussions highlighting these advantages. Evidence and References – Regarding the paragraph between lines 132–140:

- I recommend including specific references to support the claims made regarding its clinical use.

We have structured the references better in this section. Given that the validation of STAT-ON was a bit longer, we have decided to summarise and add a reference in each point.

- Although the paragraph is informative, its clarity could be improved by breaking up long sentences, avoiding extensive lists within a single sentence, and using clearer logical connectors.

We fully agree with the reviewer and have divided some sentences to improve clarity. Clinical Integration and Optimization – Suggested Comment for Lines 152–158:

- This paragraph highlights that the challenges mentioned in clinical assessments represent a significant opportunity to optimize time management during follow-up visits, which is an notable strength in improving both clinical efficiency and effectiveness. In this regard, the incorporation of specialized nursing services into movement disorders units is becoming increasingly common. Therefore, providing these professionals with training in the use of the STAT-ON device could further enhance this benefit. For example, the Hospital Vall d'Hebron has recently implemented this type of support, allowing for more effective and efficient patient care through more comprehensive pre-evaluations and better-organized neurological consultations.

We have included a paragraph in the Discussion suggesting that incorporating nursing services could further enhance the use and benefits of STAT-ON.

From Line 171 Onward – Section 2.3: Improving Quality of Life with Parkinson's Disease Therapies

- It may be useful to begin by clearly stating that there are two main types of therapies: pharmacological and non-pharmacological. This would help structure the section more clearly for the reader and emphasize that both are fundamental components of comprehensive Parkinson's disease management.

- Additionally, I would suggest that the authors include a more explicit mention of non- pharmacological options (such as physiotherapy, structured exercise, cognitive stimulation, or psychological support), which are also essential in quality-of-life-oriented care for Parkinson's disease.

- It would also be advisable to emphasize that therapeutic decisions must always be individualized, taking into account the patient's clinical characteristics, preferences, and resource availability, thereby reinforcing the patient-centered approach.

- Finally, I believe it would be highly relevant to include (ideally toward the end of the paragraph) how monitoring tools such as STAT-ON could benefit both types of therapy. While its potential has already been mentioned in relation to pharmacological treatments, it could also provide data on mobility and daily activity that may help tailor rehabilitation programs or prescribe physical exercise. Including this point could enrich the manuscript and underscore the value of

objective monitoring in clinical decision-making for both pharmacological and non- pharmacological treatments.

We thank the reviewer for these valuable suggestions. We have revised Section 2.3 to clearly distinguish pharmacological and non-pharmacological therapies, added a paragraph reviewing the benefits of non-pharmacological options for QoL, and emphasized the importance of individualized, patient-centered decisions. Finally, we included a note on how STAT-ON can support both therapy types by providing objective data on mobility and daily activity.

Research Planning and Drop-Out Reduction:

- I would suggest incorporating an important point in the manuscript regarding the potential utility of the STAT-ON device in research planning and the reduction of participant drop-out rates.

- The authors could consider the possibility that data collected in advance by the device may help optimize the planning of various clinical trials, for instance, in studies focused on physical exercise. Objective data on motor fluctuations and daily activity could contribute to better structuring of training microcycles tailored to each patient's profile, and also to the homogenization of intervention and control groups, an essential factor in clinical study design.

- Furthermore, it would be worth highlighting that the use of objective monitoring with STAT- ON during participant follow-up could yield more reliable and detailed data, thereby enhancing the quality of the resulting evidence. This tool could also help reduce drop-out rates by enabling more personalized adjustments and closer monitoring, thus improving participant adherence to research protocols.

We thank the reviewer for this insightful comment. We have added a section in the Discussion highlighting the potential applications of STAT-ON in clinical research, particularly in clinical trials as suggested. We also emphasize its value in generating more robust evidence and in reducing drop-out rates by enabling more personalized adjustments and closer monitoring.

Broader Application to Other Therapies:

- In line with the previous comment related to research, the manuscript could also consider how STAT-ON might assist in objectifying outcomes in more "subjective" therapies such as manual therapy, osteopathy, physiotherapy, and other alternative interventions.

- Having access to quantitative data on motor parameters or daily activity could provide stronger evidence of therapeutic effects and facilitate the integration of such interventions into clinical practice guided by objective outcome measures.

In line with the reviewer’s suggestion, we have expanded Section 2.3 to include both

pharmacological and non-pharmacological therapies.

Final Comment

I believe the manuscript would significantly benefit from the suggested clarifications and expansions, which would help strengthen its scientific rigo. I encourage the authors to incorporate these aspects to increase the overall impact of their work.

Response to the Editor

Dear Editor,

Thank you very much for your editorial response concerning the manuscript, PONE-D-25- 32140 entitled “Improving Parkinson’s Disease Management Through Wearable Technology: A Cost-Benefit Perspective”.

Please find enclosed the revised version to be considered for publication in `Plos One’. We are also sending a letter to the reviewers addressing their requirements. In the new version, changes and improvements have been completed.

We would like to note that some numbers in Tables 6 and 7 have been updated to reflect revised estimates of the total number of PD patients in each country. For clarity, we have also added a sentence in Section 5, Line 442, specifying the estimated patient numbers per country. This adjustment ensures that the model can be more easily understood and applied in future studies.

We hope the revised manuscript will be considered adequate. We look forward to your editorial decision.

---

## [Editor Report · Decision Letter 1]

28 Sep 2025

Improving Parkinson’s Disease Management Through Wearable Technology: A Cost-Benefit Perspective

PONE-D-25-32140R1

Dear Dr. Català,

We’re pleased to inform you that your manuscript has been judged scientifically suitable for publication and will be formally accepted for publication once it meets all outstanding technical requirements.

Kind regards,

Joel Montané, PhD

Academic Editor

PLOS ONE

---

## [Editor Report · Acceptance letter]

PONE-D-25-32140R1

PLOS ONE

Dear Dr. Català,

I'm pleased to inform you that your manuscript has been deemed suitable for publication in PLOS ONE. Congratulations! Your manuscript is now being handed over to our production team.

Kind regards,

on behalf of

Dr. Joel Montané

Academic Editor

PLOS ONE